# Two-Port “Dry Vitrectomy” as a New Surgical Technique for Rhegmatogenous Retinal Detachment: Focus on Macula-on Results

**DOI:** 10.3390/diagnostics13071301

**Published:** 2023-03-30

**Authors:** Tomaso Caporossi, Alessandra Scampoli, Filippo Tatti, Lorenzo Mangoni, Matteo Mario Carlà, Emanuele Siotto Pintor, Francesca Frongia, Claudio Iovino, Patrizio Bernardinelli, Enrico Peiretti

**Affiliations:** 1Vitreoretinal Surgery Unit, Fatebenefratelli Isola Tiberina-Gemelli Isola Hospital, 00186 Rome, Italy; 2Department of Neuroscience, Sensory Organs and Chest, Catholic University of the Sacred Heart, 00168 Rome, Italy; 3Eye Clinic, Department of Surgical Sciences, University of Cagliari, 09124 Cagliari, Italy; 4Ophthalmology Unit, Fondazione Policlinico Universitario A. Gemelli IRCCS, 00168 Rome, Italy; 5Multidisciplinary Department of Medical, Surgical and Dental Sciences, Eye Clinic, University of Campania Luigi Vanvitelli, 80138 Naples, Italy

**Keywords:** rhegmatogenous retinal detachment, vitrectomy, vitreoretinal surgery, dry vitrectomy, two-port pars plana vitrectomy

## Abstract

We evaluated a new surgical technique for treating primary rhegmatogenous retinal detachment (RRD), consisting of localized vitrectomy near the retinal break associated with drainage of subretinal fluid without infusion. Twelve eyes of twelve patients with primary RRDs with macula-on superior, temporal, and/or nasal quadrants’ RRD with retinal breaks between 8 and 4 o’clock, pseudophakic or phakic eyes, were enrolled. All eyes underwent a two-port 25-gauge vitrectomy with localized removal of the vitreous surrounding the retinal break(s), followed by a 20% SF6 injection and cryopexy. The difference between pre-operative (T0) and post-operative mean BCVA at 6 months follow-up (T6) was not statistically significant (0.16 logMAR vs. 0.21 logMAR; *p* = 0.055). Primary anatomic success at 6 months was achieved by 86% of patients. No other complications, except for two retinal re-detachments linked to an incorrect head position of the patients, were recorded. Although further studies are necessary to evaluate the treatment’s efficacy, we believe our technique could be considered a valid alternative for managing primary RRD.

## 1. Introduction

Retinal detachment (RD) is a potential cause of total and irreversible vision loss due to the separation of the neurosensory retina from the underlying retinal pigment epithelium (RPE), whose adherence is essential for normal retinal function.

Rhegmatogenous retinal detachment (RRD) is the most common type of retinal detachment, and it develops when a retinal ‘break’ allows the ingress of fluid from the vitreous cavity to the subretinal space, resulting in retinal separation [1,2,3,4,5].

The management of rhegmatogenous retinal detachments is surgical, with the aim of releasing the vitreous traction and closing the retinal break [6]. Nowadays, the available techniques are pneumatic retinopexy (PnR), scleral buckling (SB), and pars plana vitrectomy (PPV).

The term pneumoretinopexy or pneumatic retinopexy was first used by Hilton and Grizzard in 1986 to define an intervention for retinal detachment consisting of the intravitreal injection of an expandable gas and cryotherapy or photocoagulation of the retinal rupture. This procedure allows the reabsorption of subretinal fluid and the formation of a chorioretinal adhesion around the break [7].

The gas is injected into the eye using a 30G needle with the patient in the supine position. Generally, C3F8 (0.2 or 0.3 mL) or SF6 (0.5 0.8 mL) are injected, while more rarely, only 0.8 mL of filtered air may be injected [8]. The SF6 gas doubles its volume and remains inside the eye for up to 10 days, while C3F8 expands 4 times and lasts approximately 8 weeks. An anterior chamber paracentesis can be performed before or after gas injection to prevent or treat the elevation of intraocular pressure, which is harmful to the occlusion of the central retinal artery [7,8]. In terms of efficacy, recent studies show a median success rate of 69% with a range between 51% and 90% [8].

Severe intraoperative complications are rare. However, the discovery of new retinal tears or the visualization of other tears not previously identified is very common (between 12 and 23% of cases), and for this reason, it is often necessary to repeat the execution of the same procedure or other surgical techniques. The onset of epiretinal membranes, macular holes, and vitreoretinal proliferation (PVR) is also recognized among the complications of pneumoretinopexy. The latest studies have shown an incidence of PVR following pneumoretinopexy with a median of around 5% [8].

The relative simplicity of this surgical technique, its favorable results, and the low complication rate have led many authors to encourage its use for selected cases of rhegmatogenous retinal detachment. However, pneumatic retinopexy is typically used in “uncomplicated” retinal detachments with well-visualized small tears in the upper retinal quadrants (“superior 8 clock hours”), anterior to the equator, and with PVR grade less than C [8]. Moreover, many factors including lens opacity and small pupil can limit this technique due to the difficulty to identify retinal breaks preoperatively [9]. A valuable treatment option in phakic eyes with uncomplicated RRD or medium complexity RRD is represented by scleral buckling [10].

Scleral buckling is defined as the ab externo approach that consists of suturing buckling elements onto the scleral wall, without removing the vitreous gel, to reduce vitreoretinal tractions and to close retinal breaks. Crioretinopexy is performed to ensure proper chorioretinal adhesion, and the evacuative puncture is important to drain subretinal fluid intraoperatively.

Recently, a common practice has been to use an indirect noncontact wide-angle viewing system with a chandelier endoilluminator during the scleral buckling surgery to help surgeons to acquire a wide and clear view of the retinal breaks during surgery for retinal detachment [11].

Although recent studies have reported a high anatomical success (53–83%), this technique has lost popularity compared to vitrectomy because it remains a complicated surgery with a high risk of intraoperative and post-operative complications such as bleeding (e.g., choroidal/subretinal/retinal hemorrhages), vitreous/retinal incarceration, retinal perforation, and hypotony [12].

Another treatment for RRD is pars plana vitrectomy, which was introduced by Robert Machemer around the 1970s when he developed a 17G-sized vitrectomy with pars plana insertion with cutting and suction infusion together (VISC) [13]. In 1974, O’Malley and Heintz reduced the caliber to 20 G (0.9 mm), separating the infusion system from the cutting system. In 2002 Fujii, De Juan et al. presented the 25 G transconjunctival vitrectomy system (“transconjunctival sutureless vitrectomy”) with the aim of completely replacing the 20 G [14]. This system, consisting of three 25 G trocars to allow transcleral accesses (infero-temporal, superior-temporal, and super-nasal), infusion, and illumination cannulas, was later modified by some authors to use larger caliber instruments (23 G) [15]. More recently, in 2010, Yusuke Oshima instead introduced a procedure with 27 G instruments [16].

The instruments used in vitrectomy include a microcannula, which consists of a 4 mm thin tube that, once fixed to the inferotemporal trocar, allows continuous infusion inside the vitreous chamber, a vitrectomy, and an illumination probe. To these instruments is added a vast assortment of instruments, such as forceps, scissors, soft tip cannulas, laser probes, and diathermy probes.

During the traditional pars plana vitrectomy for retinal detachment, the retinal surgeon creates three transcleral accesses with 25 G valved trocars, through which he removes the traction of the vitreous on the retina, treats retinal tears with cryotherapy or laser and injects a medium buffer (gas or silicone oil) [17].

The major innovations of the 25 G system, such as the caliber of the instruments (0.54 mm), the fluidics and cannulae system, the access of the instruments, together the increased comfort for the patient, have led to a substantial decrease in the percentage of scleral buckling in favor of vitrectomy. In fact, vitrectomy is the most performed surgical procedure for rhegmatogenous retinal detachment in most countries [18]. However, there is an ongoing debate about the choice of the surgical approach for retinal detachment. It is currently a common understanding that the choice depends on various factors, such as the number, location, and size of the breaks, the surgeon’s experience, and the presence of PVR [19].

The two large comparative studies of pars plana vitrectomy with buckling (Heimann) [20] and pneumoretinopexy (Hillier) [21] showed the primary anatomic success of vitrectomy of 72% (vs. 53.4% of scleral buckling) and 93%, respectively (vs. 80.8% of Pneumoretinopexy) (Table 1).

Despite the important evolutions in the field of vitrectomy, the use of this technique remains questionable for some selected cases. Vitrectomy is, in fact, associated with a series of complications, such as iatrogenic retinal tears and increased intraocular pressure [19].

Numerous studies have demonstrated that a complete vitrectomy causes increased oxygen levels within the vitreous chamber, causing cataract progression, and that complete vitrectomy surgery with prolonged time inside the vitreous cavity promotes an inflammatory response, which represents a risk for the development of vitreoretinal proliferation [19,22,23]. For these reasons, in recent years, research has focalized on developing a less invasive procedure than traditional vitrectomy for selected cases.

The first experiences of less invasive variants of classical vitrectomy were based on the use of two transconjunctival ports (“two-port vitrectomy”), with the aid of a slit lamp as a lighting system [24] or by means of infusion cannulas combined with probes of endoillumination [25,26].

Subsequently, to reduce the invasiveness of the surgical procedure, it was decided to limit the vitrectomy near the retinal rupture with the aim of eliminating the vitreous tractions and draining the subretinal fluid (“localized vitrectomy”) [23,27]. In 2018, Bonfiglio et al. demonstrated 94% success for single surgery and 100% success after additional procedures for 32 eyes affected by retinal macular detachment ON with intermediate tears and clear vitreous traction. The technique described by the authors involved the use of three 25 G trocars, the localization of the closed infusion tears, the removal of the vitreous around the retinal rupture under air (continuous infusion at 30–35 mmHg) and the drainage of the subretinal fluid through the rupture, then treated with an endolaser. No central vitrectomy or” shaving “of the vitreous base was performed, and tamponade in all cases was with air [27].

Mura et al. described a similar technique called “minimal interface vitrectomy”, characterized by a sectorial vitrectomy (25 G) under air (continuous infusion at 35 mmHg), performed around the rupture to release the anteroposterior and tangential traction forces and to drain the subretinal fluid. Differently from the technique described above, a partial vitrectomy was performed in the center of the vitreous chamber to allow adequate tamponade with air or gas (SF6). The rupture was instead treated with endolaser or cryotreatment. From the study reported in the literature, 12 eyes had an anatomical success of 100% [23].

Our study aims to develop a novel surgical technique, such as pneumoretinopexy in terms of invasiveness and surgical timing, and similar to pars plana vitrectomy in terms of efficacy and incidence of complications. This technique is based on the use of two access ports for the light probe and the vitrectomy, without the use of a continuous infusion. The use of a light probe controlled by the surgeon and the “wide-angle viewing system” guaranteed by the microscope allow better localization of the breaks than the “two-port” techniques described above. The vitrectomy, similar to the procedures of “localized vitrectomy” and “minimal interface vitrectomy”, is instead used to remove the vitreoretinal tractions near the ruptures and to drain the subretinal fluid. Given the removal of a limited amount of vitreous, tamponade can therefore be the same as pneumoretinopexy without the need for anterior chamber paracentesis.

## 2. Materials and Methods

This was a prospective multicentric study conducted at the University Hospital of Cagliari, Italy and at IRCCS Fondazione Policlinico Universitario A. Gemelli, Rome, Italy. Twelve eyes of twelve patients were included in the study between January 2022 and June 2022. All of them underwent a complete ophthalmic examination by a retinal specialist in both centers to assess if all the inclusion criteria were met.

The patients analyzed in this work suffered from macula-on, treatment-naïve rhegmatogenous retinal detachment, which involved the superior, temporal, and/or nasal quadrants. The retinal breaks had to be in the superior meridians (between 10 o’clock and 2 o’clock) (Figure 1). They could be pseudophakic or phakic patients with a grade ≤ 2 cataract in Thompson grading [28] and in any refractive status, including high myopia. A mild localized vitreous hemorrhage was tolerated as long as the optic disk, the retinal vessels, and the whole periphery were clearly visible.

Patients with any other type of retinal detachment (exudative, tractional, or mixed) were excluded from the study.

The mean axial length was 25.9 ± 2.8 mm, the retinal detachment involved one or two quadrants (mean 1.5 ± 0.5 quadrants), 6 eyes 1 quadrant and 6 eyes 2 quadrants, the number of the retinal breaks were one in 6 eyes (50%), 2 in 4 eyes (33%) and 3 in 2 eyes (17%) (mean number of retinal breaks 1.6 ± 0.7), all the eyes were macula-on retinal detachment and in 1 eye (8.3%) there was an hemovitreous in the inferior quadrant. The mean preoperative BCVA was 0.2 ± 0.3 logMAR (range 0–1 logMAR). Preoperative mean intraocular pressure (IOP) was 15.8 ± 2 mmHg. (Table 2).

The study adhered to the declaration of Helsinki. The protocol used was approved by the local Ethical Review Board (NP/0005165/23). Informed consent was obtained before enrollment from all participants.

### 2.1. Surgical Technique

The surgeries were performed by a single surgeon for each center (E.P. and T.C.) under local anesthesia (parabulbar block by 5cc Lidocaine and 5 ccs Bupivacaine). Surgical procedures were performed using the Constellation^®^ Vitrectomy system (Alcon, Fort Worth, TX, USA) and BIOM wide-angle viewing system (Oculus Optikgeräte GmbH, Wetzlar, Germany) in Cagliari and with the Constellation^®^ and the NGENUITY^®^ 3D Visualization System (Alcon, Fort Worth, TX, USA) in Rome. After the disinfection of the surgical field with 5% povidone-iodine application for 2 min, two 25 G valved trocars were inserted at 3.5 (pseudophakic) or 4 mm (phakic) from the limbus in the superotemporal and superonasal quadrants. The detachment was then visualized using the light pipe and the wide-angle/3D visualization viewing system.

As a first step, the retinal periphery was checked to identify any rhegmatogenous lesion or secondary tear to be treated with cryo-retinopexy before vitrectomy.

Then, a localized vitrectomy close to the main retinal break was performed with the 25 G vitrectomy, under light pipe illumination, without any infusion system (cut rate 7500 cpm, maximum vacuum 650 mmHg) ( See Appendix A).

The purpose was to release any vitreal tangential traction force on the retina and to drain all the possible subretinal fluid. During the aspiration, it was crucial to assess if any hypotony occurred by refilling the vitreal chamber with 20% sulfur hexafluoride (SF_6_). Then, a criopexy treatment was performed on the main retinal tear. Once the vitrectomy procedure was completed, an additional 20% SF_6_ was injected to reach a proper filling of the eye. Surgeons had to check the ocular pressure digitally during injection and the artery perfusion at the optic disc at the end of the filling procedure. Then, patients were placed face down for 1 h after surgery to avoid any fluid slippage, and the intraocular pressure (IOP) was checked again before their discharge. Patients were instructed about the correct head positioning for the post-operative 5 days according to the localization of the breaks, and a steroid-antibiotic association drop was prescribed 4 times per day.

### 2.2. Follow-Up Visit

Follow-up was performed on day 1 and day 7 and at 1, 3, and 6 months after surgery and evaluated by a retinal specialist in both centers. On day 1, an anterior segment evaluation, IOP measurement, and indirect ophthalmoscopy were performed. In some cases, fundus photography was taken to better assess the gas filling (Figure 2).

At the following visits, the best corrected visual acuity (BCVA) was collected, and an optical coherence tomography (Spectralis OCT, Heidelberg Engineering, Heidelberg, Germany) was performed along with the other evaluations.

Surgical failure was defined by the persistence of post-operative SRF connected to the retinal breaks, which made necessary a secondary intervention to reattach the retina. The progression of cataracts within 3 months of the primary procedure, or immediate IOP increase, were considered complications of the surgery.

### 2.3. Statistical Analysis

Visual acuity values were converted in LogMAR using Excel (Microsoft, Redmond, MA, USA). An unpaired *t*-test was used to compare the mean values. *p*-values *p* < 0.05 were considered statistically significant.

## 3. Results

In this work, we analyze the results of our innovative technologies in a group of twelve eyes affected by macula-on RRD. One eye was high myopic with an axial length of 31.05 mm. The mean duration of symptom onset was 3 days, with a range of 1 to 5 days. All the patients completed the 6 months follow-up.

Ten eyes (84%) had gained a complete macular reattachment without any complication. Only two patients (16%) did not reach the main outcome of the study, which consisted of anatomical success, showing a retinal redetachment in the first month.

The difference between pre-operative (T0) and post-operative BCVA at 7 days, 1 month, 3 months, and 6 months (T6) follow-up was not statistically significant (respectively, *p* = 0.075, *p* = 0.100, *p* = 0.055 and 0.055). (Figure 3). The mean pre-operative BCVA was 20/29 Snellen (0.16 logMAR), with a range of 20/20 and 20/200 Snellen. The mean 7 days BCVA was 20/50, the 1-month BCVA was 20/40, the 3-month BCVA was 20/32, and the 6-month BCVA was 20/32 (0.21 logMAR), with a range of 20/20 and 20/200 Snellen (Figure 3).

The injection of SF_6_ 20% did not raise the IOP pressure in any patients, and the grading of cataracts did not change.

The mean preoperative IOP was 15.8 ± 2 mmHg; the mean postoperative IOP was 14.3 ± 3.9 mmHg at day 1, 16 ± 2.2 mmHg at day 7, 17.3 ± 6.9 at 1 month, 15.3 ± 2.2 at 3 months, and 16.8 ± 1.7 mmHg at 6 months.

The extension of retinal detachment was at most two quadrants, with single or double tears localized in the superior quadrants.

We have recorded two complications that occurred within the first month, all represented by the recurrence of retinal detachment. One patient was treated with standard 3-port vitrectomy with SF_6_ 20% as endotamponade, and one patient with silicon oil 1000 cts as endotamponade removed 2 months after surgery. The final reattachment rate with the two procedures was 100%. No other complications, including cystoid macular edema, occurred during the six months follow-up (Figure 4).

## 4. Discussion

Rhegmatogenous retinal detachment (RRD) is the most common type, with an incidence of about 10/100,000 [1,2], and it develops when a full-thickness defect in the neurosensory retina allows the entry of fluid from the vitreous cavity to the subretinal space, leading to a separation of the neurosensory retina from the underlying retinal pigment epithelium (RPE).

Retinal detachment itself is painless, but warning signs almost always appear before it occurs or has advanced, such as the sudden appearance of flashes of light in one or both eyes (photopsia), many floaters, blurred vision, and gradually reduced peripheral and central vision. The three different clinical forms of retinal detachment are tractional, exudative, and rhegmatogenous, but it can also be a mixture of two or more of these. Tractional retinal detachment can be secondary to fibrous tissue on the retina surface, typically seen in patients affected by proliferative diabetic retinopathy. Exudative detachment is the result of ocular inflammatory conditions or choroidal tumors that produce increased fluid flow through the subretinal space.

Retinal breaks can develop from a retinal lesion at the time of posterior vitreous detachment (PVD) or because of trauma (around 10%) and inflammation of the eye. RRD can occur at any age, although the higher incidence is estimated around 60 to 70 years old [3], similar to PVD. Consequentially, an accurate ophthalmoscopic examination is required in patients with symptoms and signs of acute PVD that are at risk of immediate progression to RRD, particularly in the presence of retinal degenerations, as a lattice-like degeneration. Otherwise, a lower risk of RRD is reported in patients with chronic PVD and predisposing retinal lesions who have not immediately progressed to retinal breaks.

Numerous studies report a higher prevalence of RRD in myopic eyes, also including idiopathic macular holes in highly myopic eyes or in eyes with retinoschisis and other degenerative retinal conditions. Furthermore, previous cataract surgery can improve the risk of progression in RRD, particularly after the rupture of the posterior capsule, when it increases by 15 to 20 times [3].

The fellow eye in people with an RRD has a higher risk, with 2% to 10% of RRDs being bilateral. In at least half of fellow eyes, RRD can develop from ophthalmoscopically normal areas of the retina, while in a lower percentage, it can occur from pre-existing retinal lesions.

An accurate ophthalmoscopic examination allows the correct diagnosis of RRD. Acute RRD has the aspect of an edematous folded retina that loses its transparency and takes a bullous configuration, moving with the eye movement. Other signs of retinal detachment can be represented by vitreous hemorrhage or the appearance of “Tobacco dust”, which describes pigment cells in the anterior vitreous visible on slit lamp biomicroscopy [4]. Subretinal fibrosis and retinal cyst formation are signs of chronic retinal detachments.

The macula is still attached in ‘macula-on’ retinal detachments, compared to ‘macula-off’ RRD, which is the reason why they are treated as an emergency to guarantee a final good visual acuity and a better prognosis. In fact, in 90% of well-performed macula-on retinal detachments, vision is 20/40 or better. Moreover, about 70% to 90% of macula-on eyes reach a successful anatomical outcome after just one surgery [3]. Nevertheless, even after well-performed operations, the visual acuity can remain quite low due to macular epiretinal membranes, cystoid macular edema, and foveal photoreceptor degeneration. On the other side, new or missed retinal breaks and proliferative vitreoretinopathy (PVR) can represent the reasons for secondary retinal redetachment [5].

The management of rhegmatogenous retinal detachments is surgical, and the available techniques are pneumatic retinopexy, scleral buckling, and pars plana vitrectomy [7,8,10,14,15,16].

Pars plana vitrectomy was first introduced around the 1970s by Robert Machemer, who developed a 17G-sized vitrectomy with cutting and suction infusion together [13]. Since that time, numerous changes have been made to develop the current technology. Nowadays, vitrectomy is the most performed surgical procedure for rhegmatogenous retinal detachment [18].

Several studies have demonstrated that a complete vitrectomy causes increased oxygen levels within the vitreous chamber, causing cataract progression and that complete vitrectomy surgery with prolonged time inside the vitreous cavity promotes an inflammatory response, which represents a risk for the development of vitreoretinal proliferation [19,22,23]. For these reasons, in recent years, research has focalized on developing a less invasive procedure than traditional vitrectomy for selected cases.

The aim of our study is to develop a novel surgical technique for the management of an RRD, similar to pneumoretinopexy in terms of invasiveness and surgical timing and similar to three-ports pars plana vitrectomy in terms of efficacy and incidence of complications. Our minimally invasive technique is based on the use of two access ports for the light probe and the vitrectomy, without the use of a continuous infusion, for a localized vitrectomy close to the causative retinal break and drainage of the subretinal fluid associated with a cryo-retinopexy.

Compared to the pneumoretinopexy, where the discovery of new retinal tears or the visualization of other tears not previously identified is very common (between 12 and 23% of cases) [8] and represents the cause of surgical failure, our technique allows us to visualize all the retinal tears directly. Moreover, the stretching of the vitreous from the growing gas bubble and the limited space in the vitreous cavity create traction in other areas of the retina, leading to new breaks after pneumatic retinopexy. In terms of efficacy, recent studies have shown a median success rate of 69% with a range between 51 and 90% for pneumoretinopexy [8], while our minimal vitrectomy has achieved anatomical success in 85% of patients. Furthermore, we have also treated retinal detachments located in nasal and temporal quadrants, which represent exclusion criteria for pneumoretinopexy. A similar limit of these two procedures is represented by the importance of a correct post-operative head position. In fact, we believe that in both cases treated with our two-port dry vitrectomy, the surgical failure was associated with low compliance of the patients in maintaining the indicated post-operative head position.

Compared to scleral buckling, although recent studies have reported a high anatomical success (53–83%), it remains a complicated surgery with a high risk of intraoperative and post-operative complications [19,20].

Compared to the standard PPV, our procedure has several advantages.

First, it has reduced surgical time. Our two-port vitrectomy is rapid and uses fewer incisions and minimal vitreal manipulation with a subsequent reduction in the inflammatory response. Consequentially, no vitreoretinal proliferation (PVR) or anterior chamber inflammation has developed in any of our patients during follow-up.

Similarly, considering all 12 eyes of our study, no post-operative macular OCT has shown the onset of cystoid macular edema (CME), which represents a common complication in standard PPV. In fact, recent studies have highlighted that the incidence of CME after rhegmatogenous retinal detachment repair was higher in patients who underwent PPV, either alone or combined with phacoemulsification than in those treated with SB [29]. Probably, the use of cryotherapy retinopexy instead of endolaser retinopexy [30] plays a role in lowering the risk of post-operative CME.

Furthermore, considering that this transparent crosslinked hydrogel plays an important role in mechanical and molecular homeostasis of the eye [31], a limited vitreous removal with preservation of the cortex which protects the crystalline lens from excessive oxygen exposure reduces cataract formation. In fact, in our population, no opacification of the lens has been observed during the follow-up.

Compared with the other mini-invasive vitrectomy procedures [23,24,25,26,27], the innovation of our two-port-minimally invasive vitrectomy is to remove the vitreous surrounding the retinal breaks without infusion and vitreal hydration. Consequentially, we preserve the vitreal gel with its structure and antioxidant properties to reduce the onset of post-operative inflammation, and simultaneously, we create a space for the gas to expand, reducing secondary tractions on other areas of the retina.

## 5. Conclusions

Our study describes a new surgical procedure that integrates the advantages of existing procedures. Our aim is to preserve the vitreous body, which inhibits inflammation and cataract progression, such as scleral buckling and pneumatic retinopexy, while at the same time obtaining the advantages of PPV in terms of relieving vitreous tractions and visualizing the entire retinal periphery. Furthermore, we aim to minimize invasiveness using only two ports and shorten the total surgical time. In our technique, a limited amount of vitreous is removed, eliminating the need for any dynamic infusion of either air or fluid and the tamponade can be injected later to replace the small amount of aspirated vitreous.

This new technique is fascinating because of its reduced invasiveness, which could save time and increase safety, leading to a good functional and anatomical success rate in our initial experience with macula-on RRD. Twelve eyes is a small number, and further prospective studies, with a larger sample of patients, are needed to define the role of this novel technique.

## Figures and Tables

**Figure 1 diagnostics-13-01301-f001:**
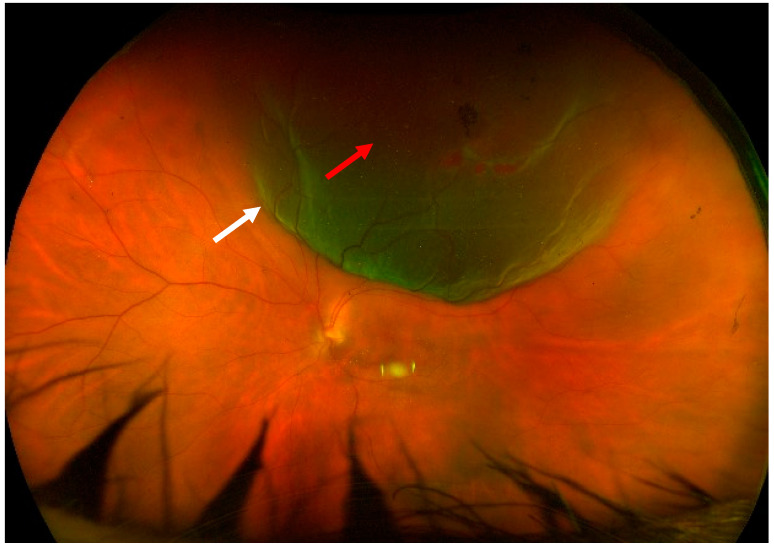
Pre-operative fundus picture. Left eye, superior RRD (white arrow) with a large retinal break at 12–1 o’clock (red arrow).

**Figure 2 diagnostics-13-01301-f002:**
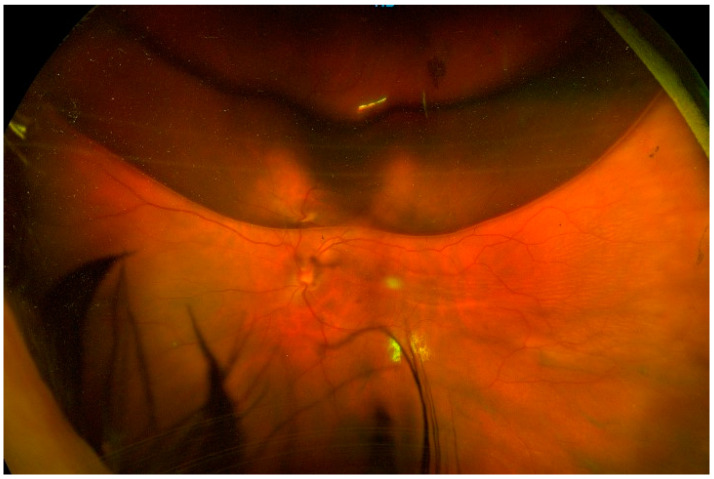
Post-operative day 1. The retinography shows a 40% filling of gas.

**Figure 3 diagnostics-13-01301-f003:**
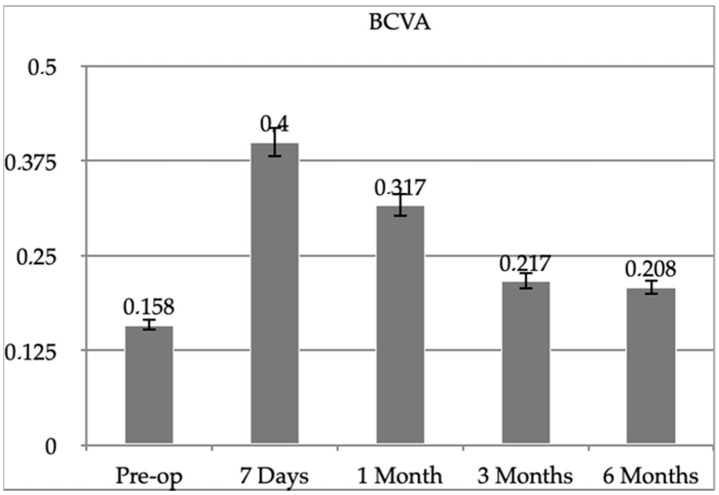
BCVA results.

**Figure 4 diagnostics-13-01301-f004:**
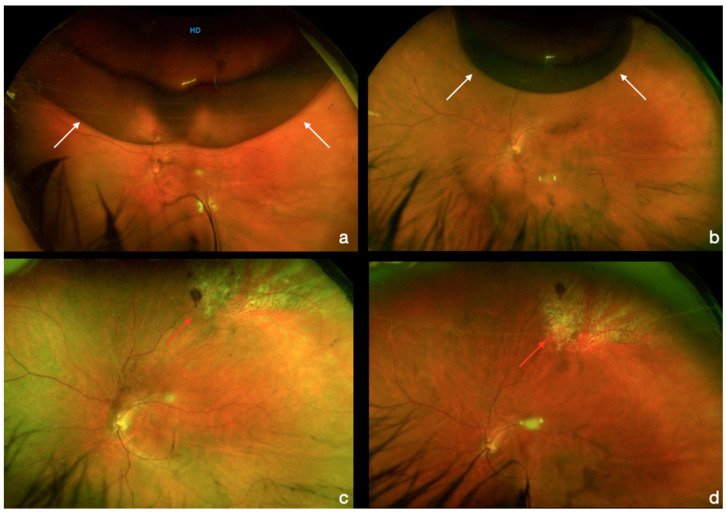
Post-operative day 1 (**a**) and day 7 (**b**) fundus pictures with a progressive reduction in gas filling (white arrows), and fundus pictures at 1 month (**c**) and 6 months (**d**) with cryopexy treatment on the main retinal tear and the retina attached (red arrows).

**Table 1 diagnostics-13-01301-t001:** Advantages and disadvantages of scleral buckling (SB), pars plana vitrectomy (PPV), and pneumatic retinopexy (PnR).

	PnR	SB	PPV
Advantages	Better post-operative VA, cost savings, less cataract progression	High reattachment rate with VA improvement in phakic eyes	High reattachment rate in pseudophakic eyes
Disadvantages	Lower reattachment rate in RRD with an inferior tear or pre-operative pseudophakia	Post-operative cataract progression	Post-operative PVR and cataract progression
Suggested use	RRD patients with pre-operative phakia, clear vitreous, retinal breaks in the superior region	Young RRD patients with pre-operative phakia	Older RRD patients with retinal tears in various regions, including GRT and PVR. Should be combined with cataract surgery in older patients

**Table 2 diagnostics-13-01301-t002:** Demographics data.

Patient ID/Age (YO)/Sex	Eye	Symptoms Onset (Days)	Axial Length (mm)	Retinal Detachment Extension (Quadrant)	Retinal Detachment Position	Haemovitreus	Retinal Breaks (Number)	Retinal Breaks (Position-Clock Hours)	Result
1/55/F	left	30	23.60	2	supero-nasal	no	3	X–XI	FAV
2/52/M	right	3	26.40	1	superior	no	2	XI–I	UNFAV
3/29/M	left	4	31.05	1	superior	no	2	XII	FAV
4/70/F	right	3	23.00	2	supero-nasal	no	1	XII	FAV
5/52/F	left	30	25.40	2	supero-temporal	no	1	XII–I	FAV
6/64/M	right	1	26.50	1	temporal	no	1	IX	FAV
7/73/F	left	3	23.95	2	supero-temporal	yes	1	I	FAV
8/61/M	left	1	26.65	1	superior	no	1	XII	FAV
9/48/M	right	5	23.91	1	nasal	no	1	IX	FAV
10/33/F	left	1	25.05	2	temporal	no	2	III–IV	FAV
11/70/M	left	1	26.50	1	temporal	no	2	V–VI	UNFAV
12/53/F	right	1	23.08	2	supero-nasal	no	2	XII–V	FAV

## Data Availability

Data is unavailable due to privacy or ethical restrictions.

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
