# Peer review of "Two-Port “Dry Vitrectomy” as a New Surgical Technique for Rhegmatogenous Retinal Detachment: Focus on Macula-on Results"

_diagnostics, 2023, doi:10.3390/diagnostics13071301_

Round 1
Reviewer 1 Report
Caporossi et al. evaluated a new surgical technique for treating primary rhegmatogenous retinal detachment. This technique consisted of a localized vitrectomy near the retinal break associated with drainage of subretinal fluid without infusion. The researchers found that this technique was effective in treating primary rhegmatogenous retinal detachment. Twelve individuals participated in the study. Primary rhegmatogenous retinal detachments with macula on, RRD in the superior, temporal, and/or nasal quadrants with retinal breaks between 8 and 4 o'clock, pseudophakic, or phakic eyes were the inclusion criteria. In each eye, a two-port 25-gauge vitrectomy was performed, during which the vitreous was removed locally from the region around the retinal break (or breaks), and then a 20% SF6 injection and cryopexy were performed. There was not a statistically significant change seen between pre-operative (T0) and post-operative (T6) mean BCVA at six months follow-up (T6) (0.16 logMAR vs 0.21 logMAR; p=0.055). At the six-month mark, primary anatomic success had been attained by 86% of patients. There were no further issues that were noted, with the exception of two retinal detachments that were caused by the patients holding their heads in the wrong posture. Although more research is required to determine whether or not the therapy is effective, we feel that our method has the potential to be regarded as a legitimate option for the management of primary RRD.
The paper is well-drafted, and the analyses have been performed excellently. However, I have the following queries that should be taken by the authors.
1. The standard deviation should be included in Figure 3, considering the results of all 12 participants.
2. Figure 4 should contain pictographs of pre-op, 7 days, 1 month, 3 months, and 6 months.
3. The retinal images and their characteristics should be labeled properly. Many readers would not be experts in the field of retinoscopy. Proper labeling would improve the understanding capability of a wider readership base.
Author Response
1)thank you for your comment. We added the SD to the graph, as you suggested.
2) Now we show better the postoperative results in figure 4
3) thank you for your comment. Now all the figure has been appropriately labelled.
Reviewer 2 Report
The study “Two-Port Dry Vitrectomy as a new surgical technique for
Rhegmatogenous Retinal Detachment: focus on macula on results” describes a new method for the treatment of retinal detachment. The authors propose a pars plana vitretomy without infusion system, combined with retinopexy and SF6-gas injection.
They report a success rate of 86% after 6 months and a success rate of 100% in 16 patients.
The study was described as a prospective study without reference to ethical approval. Why were macula-on patients chosen for the study?
The introduction could be reduced by 60-70%, the literature could be referred to in the discussion-section.
The proposed method provides a new approach. It would be important to address the advantages as well as the disadvantages, including possible complications. What are the shortcomings of the study? There is little information about preoperative characteristics of the included patients. Sixteen patients were included, this is a small number and should be mentioned.
The paper uses abbreviations - some of them were not explained (“SB” and “PnR” (l.388)).
Author Response
1) Thank you for your comment. We have written about the ethical approval "The protocol used was approved by the local Ethical Review Board (NP/0005165/23). " We have decided to analyze macula on patients because of their higher necessity to reduce the rate of complications and the time of surgery for a better prognosis.
2) thank you for your comment. The introduction has been improved to reach the amount of 4000 world requests as the minimum for the publication. The references have been added in the discussion section.
3) thank you for your comment. Now all the preoperative characteristics of the population have been improved to understand better. Line 241-246. this is a pilot study, and the population is small, as mentioned at the end of the discussion section.
4) thank you for your comment. All the abbreviations are now fixed.
Reviewer 3 Report
In this study, Caporossi et al. describe a surgical method for fovea-on rhegmatogenous retinal detachment. This is a small study of a surgical method with 6-months results. Such studies are important for the advancement of our clinical practice.
1. General comment: This study is marked by poor English language. A lot of grammatical errors make this study difficult to read. Please either seek help for profession language revision, otherwise the journal/MDPI may provide assistance for language revision.
2. Table 1. This is currently formated as a screenshot of a table. Please note the red lines from the language recognition software. Instead of this figure/pseudo-table, please provide a table with correct formatting.
3. Methods. Did the authors perform this surgery in cases prior to this, or are these the only cases available? I would expect that the first cases would have a certain level of learning/familiarity aspect and hence there would be a higher risk of post-operative complications. This is an aspect that should be elaborated in the methods and discussed later in the paper.
4. Results. For IOP data, there is only a conclusion. Instead, can you provide actual IOP data?
5. Results, line 314-316: This sentence associating postoperative head position to surgical failure is purely speculative. This is not a result, as you have not studied this aspect in a valid manner. Please omit this sentence. You may comment that this may be related to postoperative head position in the discussion, but strictly speaking, you cannot be sure of the causality of this.
6. Discussion, line 394. Please refrain from calling this method for non-invasive. I understand your underlying reasoning, this surgical procedure provides less alteration. But this is surgery and is, by definition, invasive.
Author Response
1) thank you for your comment. An English editor has revised the text.
2) thank you for your comment. The table is now uploaded in the correct format
3) thank you for your comment. Now the materials and method section has been implemented with the characteristics of the population of the study. We have included all the macula-on patients we initially treated. line 241
4) thank you for your comment. All the data of the IOP have been added, as you suggested. line 323
5) thank you for your comment; we have written about the head position in the discussion.
6) thank you for your comment. We have changed the sentence from noninvasiveness to "reduced invasiveness." as you suggested.
Round 2
Reviewer 1 Report
May be accepted
Author Response
Thank you for your appreciation
Reviewer 2 Report
In their revision, the authors have added ethical approval by the local Ethical Review Board.
The introduction follows the requested word count. It would have been nice to learn about the relevance of the infusion port, as this is the missing component of the proposed method.
The preoperative characteristics of the population were added and imply that there were myopic patients included. It would have been interesting to include the characteristics of the patients, the size and location of the breaks in the main paper, as these factors contribute to the success rate of the procedure.
Author Response
1)thank you for your comment. We have explained why we don't use the infusion port in the final part of the discussion. Line 397-402.
2) thank you for your comment. as you suggested, we added a table (table 2) to describe the single patient characteristics and the relations with the retinal reattachment results in the materials and method section.